# Source identification and distribution reveals the potential of the geochemical Antarctic sea ice proxy IPSO$_{25}$

S.T. Belt[1], L. Smik[1], T.A. Brown[1], J.-H. Kim[2], S.J. Rowland[1], C.S. Allen[3], J.-K. Gal[2], K.-H. Shin[2], J.I. Lee[4] & K.W.R. Taylor[5]

The presence of a di-unsaturated highly branched isoprenoid (HBI) lipid biomarker (diene II) in Southern Ocean sediments has previously been proposed as a proxy measure of palaeo Antarctic sea ice. Here we show that a source of diene II is the sympagic diatom *Berkeleya adeliensis* Medlin. Furthermore, the propensity for *B. adeliensis* to flourish in platelet ice is reflected by an offshore downward gradient in diene II concentration in >100 surface sediments from Antarctic coastal and near-coastal environments. Since platelet ice formation is strongly associated with super-cooled freshwater inflow, we further hypothesize that sedimentary diene II provides a potentially sensitive proxy indicator of landfast sea ice influenced by meltwater discharge from nearby glaciers and ice shelves, and re-examination of some previous diene II downcore records supports this hypothesis. The term IPSO$_{25}$—Ice Proxy for the Southern Ocean with 25 carbon atoms—is proposed as a proxy name for diene II.

[1] School of Geography, Earth and Environmental Sciences, University of Plymouth, Plymouth, PL4 8AA UK. [2] Department of Marine Science and Convergence Technology, Hanyang University ERICA Campus, 55 Hanyangdaehak-ro, Sangnok-gu, Ansan-si, Gyeonggi-do 426-791, South Korea. [3] British Antarctic Survey, High Cross, Madingley Road, Cambridge CB3 0ET, UK. [4] Korea Polar Research Institute, 26 Songdomirae-ro, Yeonsu-gu, Incheon 21990, South Korea. [5] Isoprime Limited, Isoprime House, Earl Road, Cheadle Hulme, Stockport SK8 6PT, UK. Correspondence and requests for materials should be addressed to S.T.B. (email: sbelt@plymouth.ac.uk).

Sea ice in the Southern Ocean is one of the most seasonal and variable features of the Earth's surface, and has a significant influence on key oceanic and atmospheric processes, which, in turn, have major impacts on global climate[1]. Although Antarctic sea ice extent has undergone a slight overall increase in recent decades, this is not the case for all regions, with dramatic reductions in the Bellingshausen and Amundsen Seas, being of particular note[2]. Reconstruction of sea ice conditions over longer timeframes is, therefore, critical for contextualizing recent changes, and for the broader interpretation of past climate conditions and the prediction of future climate states[3]. Some models, for example, suggest that Antarctic sea ice extent will reduce by 24% and more than a third in terms of total volume by 2100 (refs 3,4), while others have predicted shorter sea ice seasons[5]. Currently, however, the general paucity of palaeo sea ice records from the Southern Ocean impacts on modern climate modelling[6], and further reconstructions are needed to better inform, and test, current modelling efforts. From a methodological perspective, analysis of fossil diatoms archived in marine sediments forms the basis of established approaches for the determination of Antarctic winter and summer sea ice extent, although these become less reliable beyond the late Quaternary as extinct taxa with uncertain ecological affiliations become more prevalent[7]. Further, confidence in the reconstruction of past Antarctic summer sea ice extent is weaker than for winter sea ice due to fewer reference or analogue samples, although its determination represents an essential parameter with respect to evaluating seasonal sea ice cycles and net changes[8,9].

In recent years, the analysis of an organic geochemical lipid biomarker—a so-called highly branched isoprenoid (HBI) diene (until now referred to as diene II; Fig. 1)—has been proposed as a possible proxy measure of Antarctic sea ice[10–18]. Diene II is a close structural analogue of the mono-unsaturated HBI lipid $IP_{25}$, which has become a well-established proxy for seasonal sea ice in the Arctic[19,20]. In fact, diene II—hereafter referred to as $IPSO_{25}$: Ice Proxy for the Southern Ocean with 25 carbon atoms, by analogy with $IP_{25}$ for the Arctic[19]—co-occurs with $IP_{25}$ in certain Arctic sea ice diatoms[21] and its distribution in Arctic sediments exhibits a close parallel to that of $IP_{25}$ (ref. 20). However, $IP_{25}$ has not been reported in the Antarctic, likely due to the absence of the specific diatoms that biosynthesize this biomarker in the Southern Ocean. In contrast, $IPSO_{25}$ has been reported in Antarctic sea ice and near-surface sediments, albeit in a relatively small number of studies[12,22–24], with a stable isotopic composition ($\delta^{13}C$ = ca. − 5 to − 18‰) indicative of a sea ice diatom origin[12,24]. Importantly though, unlike $IP_{25}$, for which the source[21], seasonal production[25] and distribution pattern across the Arctic have been determined following analysis of sea ice and several hundred surface sediments from different Arctic regions[20,26], no source has been identified for $IPSO_{25}$. As such, although a number of sea ice reconstructions based on $IPSO_{25}$ have been reported, spanning recent decades, the Holocene[10–14,16,17] and the last glacial (ca. 60 kyr BP (ref. 15)), the lack of knowledge of its source, has almost certainly had an impact on the interpretation of its sedimentary occurrence and abundance characteristics.

In the current study, we identified $IPSO_{25}$ in a single sea ice endemic (sympagic) diatom species by manual isolation of individual cells from a bulk sample of mixed sea ice diatom assemblage collected from the West Antarctic Peninsula (WAP) in December 2014, and analysis of its lipid composition by gas chromatography–mass spectrometry (GC–MS). We also determined the abundance distribution of $IPSO_{25}$ in ca. 150

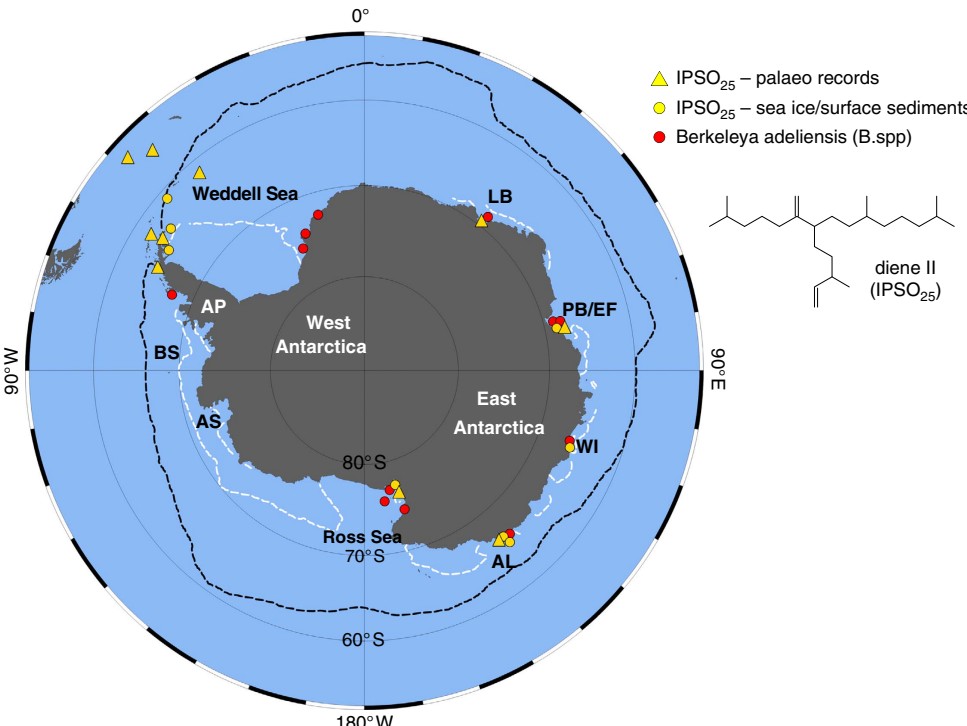

**Figure 1 | Distribution of *Berkeleya adeliensis* and IPSO$_{25}$ in previous studies.** Summary map of Antarctica showing locations where *Berkeleya adeliensis* and HBI diene II (IPSO$_{25}$) have been identified in sea ice, surface sediments and palaeo records (note: *B*. spp designates studies where diatom identification was not conducted beyond the genus level). The black and white stippled lines refer to the median winter and summer sea ice margins for the interval 1979–2010 (National Snow and Ice Data Center), respectively. Abbreviated location names are as follows: AP, Antarctic Peninsula; BS, Bellingshausen Sea; AS, Amundsen Sea; AL, Adélie Land; WI, Windmill Islands; PB/EF, Prydz Bay/Ellis Fjord; LB, Lützow-Holm Bay. The structure of the di-unsaturated HBI lipid biomarker diene II (IPSO$_{25}$) is also shown.

surface sediments covering different regions of (mainly) West Antarctica, including the Weddell Sea, the Antarctic Peninsula (AP), the Bellingshausen Sea and the Ross Sea, together with its stable isotopic composition ($\delta^{13}$C) in three representative surface sediments from some of these locations. The source of IPSO$_{25}$ in our sea ice samples, *Berkeleya adeliensis* Medlin[27], is a widespread and commonly occurring constituent of Antarctic sea ice; both important attributes for paleo sea ice reconstruction purposes, yet this diatom is rarely evident in fossil assemblages. As a tube-dwelling species, *B. adeliensis* is particularly well adapted for growth in the relatively open channels of platelet ice, a common feature of landfast ice proximal to coastal Antarctic locations, which probably has important consequences for the interpretation of sedimentary IPSO$_{25}$.

## Results

**IPSO$_{25}$ in sea ice and picked sea ice diatom cells.** The diatom assemblage in a bulk sea ice sample collected from the WAP in December 2014 consisted of *B. adeliensis* as the major diatom species (>50%) (Fig. 2). Further, the lipid biomarker IPSO$_{25}$ was identified in non-polar extracts obtained from both the bulk sea

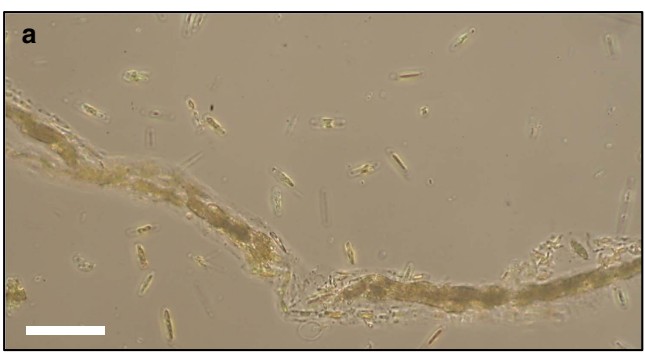

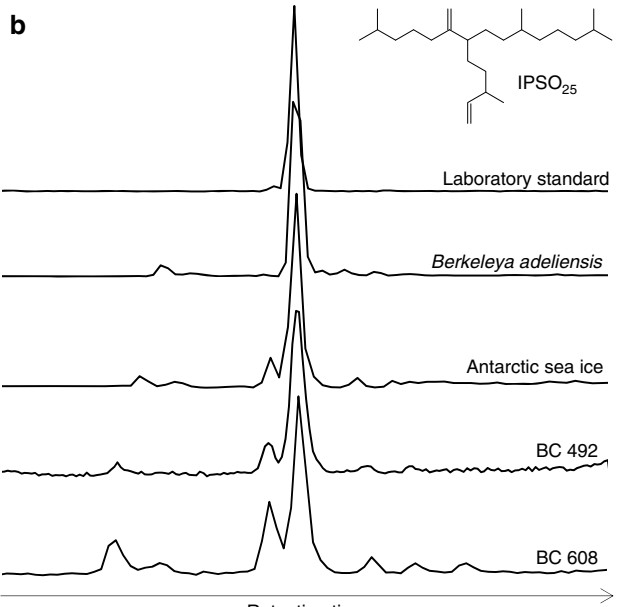

**Figure 2 | IPSO$_{25}$-producing species and lipid extracts containing IPSO$_{25}$.** (**a**) Light microscopy image of *B. adeliensis* isolated from a sample of Antarctic sea ice immediately following partial thawing. The strands of cells are particularly characteristic of this tube-dwelling species. The white scale bar represents 100 μm; (**b**) partial GC–MS chromatograms of IPSO$_{25}$ in various samples (SIM mode; *m/z* 348.3). BC492 and BC608 are surface sediments. The structure of IPSO$_{25}$ is also shown.

ice sample and picked cells of *B. adeliensis*, with an estimated cellular concentration (ca. 6 pg per cell) consistent with that of other HBI-producing diatoms[21]. No other HBI lipids were identified in the bulk sea ice sample or the picked cells of *B. adeliensis*.

**IPSO$_{25}$ in Antarctic surface sediments.** Within the surface sediments, IPSO$_{25}$ could be identified and quantified in 125 out of the 149 samples analysed, with concentrations ranging from 0.22 to 1,830 ng g$^{-1}$ (Fig. 3; Supplementary Table 1). Highest concentrations (>500 ng g$^{-1}$) were always associated with samples taken from coastal locations, with a general drop-off in abundance for offshore sites (Fig. 4). In some cases, there were also substantial abundance variations in IPSO$_{25}$, even for relatively nearby study sites. Finally, the stable isotopic composition ($\delta^{13}$C) of IPSO$_{25}$ in surface sediments from Marguerite Bay (WAP), the SE Weddell Sea and the northern AP was found to be $-15.00 \pm 0.03$‰, $-13.46 \pm 0.02$‰ and $-14.39 \pm 0.07$‰, respectively (Supplementary Fig. 1).

## Discussion

The data presented herein establish *B. adeliensis* as a source of IPSO$_{25}$ in Antarctic sea ice. Similar HBI alkenes have recently been identified in a different species within the genus *Berkeleya* (*B. rutilans*[28]), including an HBI diene, but this was a different isomer to IPSO$_{25}$ identified here in Antarctic sea ice and sediments. Determining whether *B. adeliensis* represents the only source of IPSO$_{25}$ in Antarctic sea ice will require analysis of samples from other Antarctic regions, and individual diatom species within these. In the meantime, we note that, among the more common and abundant Antarctic sea ice diatom genera, *Fragilariopsis*, *Chaetoceros* and *Nitzschia*, are not producers of HBI lipids[29]. In fact, of the known HBI-producing diatom genera, only *Haslea*, *Pleurosigma*, *Navicula* and *Berkeleya* are found in Antarctic sea ice, and only the latter two contain species that are generally considered to be common and abundant (*viz. Navicula glaciei* and *B. adeliensis*); however, no HBI lipids (including IPSO$_{25}$) were identified previously in *N. glaciei*[12]. In addition, species within the *Pleurosigma* genus do not biosynthesize HBIs with a 6–17 double bond[30], which is one of the structural characteristics of IPSO$_{25}$ (Fig. 2). In previous studies of HBI lipids in Antarctic sea ice and sea ice diatoms[12,22,23], only IPSO$_{25}$ has been identified as a common and abundant component, despite analyses having been conducted on mixed diatom assemblages of varying composition. This either means that *B. adeliensis* is a unique source of IPSO$_{25}$ in Antarctic sea ice or that other species also only produce this particular HBI, and no others. However, the latter seems unlikely given the large number of HBIs of different structural type made by various diatoms and, unlike all other HBI-producing diatom genera, which have thus far, always been shown to produce a suite of HBIs (typically >3) (for a recent review, see ref. 28 and references therein), the genus *Berkeleya* appears to biosynthesize predominantly only one isomer. Further, the occurrence of up to seven HBI isomers in Arctic sea ice[31] can be attributed to the presence of several HBI-producing diatoms, and not just those that biosynthesize IP$_{25}$ (refs 19,21,25). Therefore, although it is feasible that species other than *B. adeliensis* may be producers of IPSO$_{25}$ in Antarctic sea ice, these are likely to be only relatively minor contributors of this biomarker. As such, we believe that *B. adeliensis* probably represents a major source of IPSO$_{25}$ in coastal Antarctic sea ice and underlying sediments, and further data support this hypothesis. For example, the identification of IPSO$_{25}$ in sediments from a large number of coastal locations is consistent with the presence of *B. adeliensis* as a common and

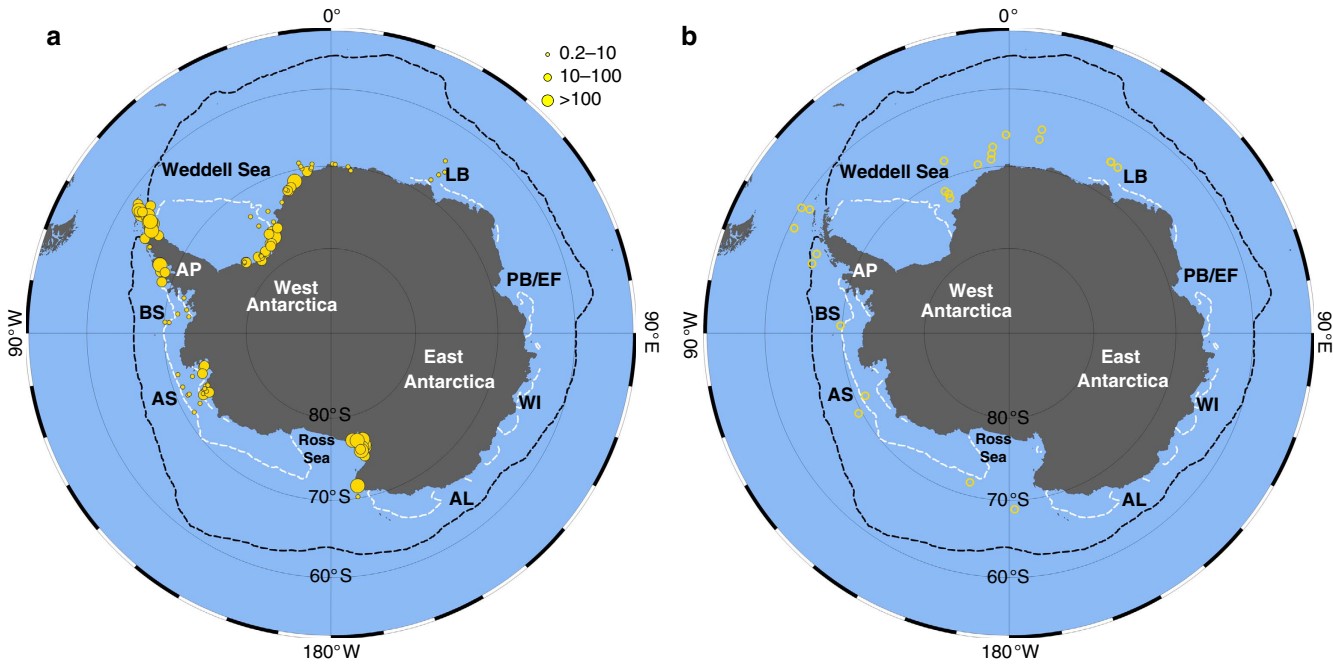

**Figure 3 | IPSO$_{25}$ in surface sediments. (a)** Distribution map showing the variable concentration of IPSO$_{25}$ in Antarctic surface sediments (note: the key refers to this map only and values correspond to concentration ranges of IPSO$_{25}$ in ng g$^{-1}$ dry sediment); **(b)** Locations where IPSO$_{25}$ was below the limit of detection (that is, < 0.2 ng g$^{-1}$ dry sediment).

relatively abundant species within Antarctic landfast ice diatom communities[27,32–34]. On a more specific regional basis, IPSO$_{25}$ has previously been reported in sea ice and/or sediments from locations where *B. adeliensis* has also been identified in sea ice, including Ellis Fjord[24,35], Adélie Land[11,12,18,34,36], Lützow-Holm Bay[37,38], the WAP[13,16] (and this study) and the Ross Sea/McMurdo Sound[23,33,39,40] (Fig. 1). For the Windmill Islands, East Antarctica, *B. adeliensis* has also been identified in sediments from Stephenson Cove[41], while IPSO$_{25}$ has been reported in sea ice from the same location and nearby O'Brien Bay[12] (Fig. 1).

*B. adeliensis* is a constituent species of the tube-dwelling genus *Berkeleya* Grunow, found frequently in Antarctic coastal waters and sea ice[33,34,36,42,43], and is commonly associated with landfast ice, where it can colonize both consolidated bottom ice and platelet ice; in some cases forming elongated strands that extend below the ice surface[34,36,42,43]. On the other hand, *B. adeliensis* has not been reported outside of the Antarctic, so cannot be a source of IPSO$_{25}$ in sediments from the Arctic, for example, despite the common occurrence of this biomarker in sediments from high latitude northern hemisphere settings[20]. A more specific habitat preference for *B. adeliensis* in Antarctic sea ice is not entirely clear, since elevated cell numbers have been reported in bottom ice and platelet ice[34,36], although some migration to the latter has been noted during ice melt[36]. Indeed, Riaux-Gobin *et al.*[36] suggested that *B. adeliensis* may be more tolerant towards the contrasting environments of bottom ice and platelet ice than many other sympagic species, with the more open-channel network of platelet ice possibly more compatible with such tube-dwelling species[34]. Further, Riaux-Gobin *et al.*[34] identified substantially higher abundances of *B. adeliensis* in sea ice from coastal locations compared with offshore settings around Adélie Land, East Antarctica, while in the water column, generally low abundances of *B. adeliensis*, attributed to ice melt release, were restricted to under-ice waters or polynyas proximal to coastal locations[34,43].

The close association of *B. adeliensis* with landfast ice prompted Riaux-Gobin *et al.*[43] to suggest that the occurrence of the cells of such sympagic species may represent suitable proxies in sedimentary records, adding the caveat that the poor preservation potential of the cells may prevent such an application, in practice. Indeed, *B. adeliensis* was not identified in sediment assemblage counts from Lützow-Holm Bay, despite being abundant in overlying sea ice and the water column[37], and the organism is rarely (if ever) a constituent of diatom inventories used in paleoceanographic reconstructions[9,44–46]. Abundances of *B. adeliensis* have been shown to exhibit a high-to-low abundance trend for coastal to offshore sites for Adélie Land in both sea ice[34] and surface waters soon after ice melt[43], and a similar trend in IPSO$_{25}$ distribution has been observed previously in surface sediments from the same region, with highest concentrations for near-shore settings, lower values offshore, and absence for sites beyond the marginal ice zone[12]. A similar offshore depletion in abundances of IPSO$_{25}$ was also identified previously in sediments from Lützow-Holm Bay[38], from where *B. adeliensis* has been reported in coastal landfast ice[37] and the generality of this gradient can be readily seen through inspection of the surface sediment data for IPSO$_{25}$ presented herein, with highest concentrations (up to ca. 1,800 ng g$^{-1}$) for coastal locations and substantially lower abundances offshore (ca. 0.22–8.51 ng g$^{-1}$) (Fig. 4). Finally, concentrations of IPSO$_{25}$ in surface waters from East Antarctica during late spring (that is, during and shortly after ice melt) were recently shown to be strongly dependent on the nature and length of seasonal ice cover, with highest values for coastal locations experiencing at least partial ice cover extending into the summer[47].

Collectively, these observations suggest that distributions of IPSO$_{25}$ in Antarctic sediments are closely related to the ecology of the source diatom, *B. adeliensis*. If this is the case, the apparently strict association of *B. adeliensis* to landfast ice might limit the value of IPSO$_{25}$ as a general Antarctic sea ice proxy. However, instead, the environmental specificity of its source should enable more detailed insights into defining past sea ice conditions to be made with confidence. Thus, since it has been shown previously that *B. adeliensis* increases in abundance towards the end of the spring bloom and during the onset of ice melt[34,40], it follows that

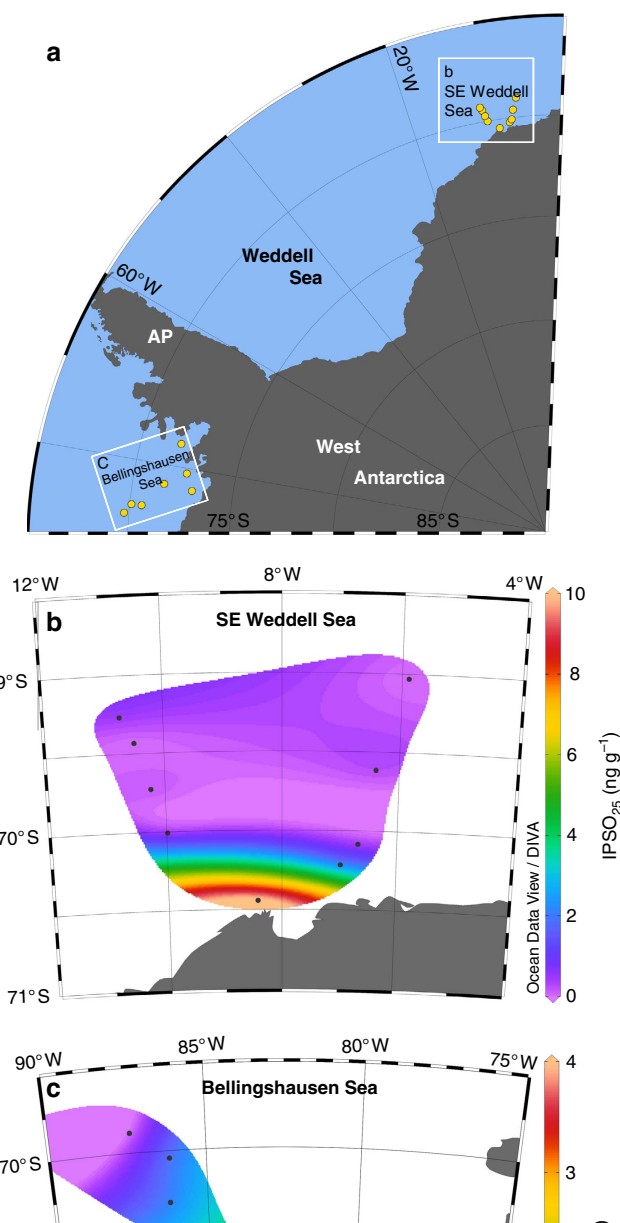

**Figure 4 | IPSO$_{25}$ distribution in coastal-offshore transects.** Regional maps illustrating the gradient drop-off in IPSO$_{25}$ concentration in surface sediments from coastal to offshore settings. (**a**) Map of West Antarctica showing two named sub-regions and individual sampling locations for which IPSO$_{25}$ data are further illustrated in **b** and **c**. (**b**) IPSO$_{25}$ concentrations in surface sediments from the SE Weddell Sea; (**c**) IPSO$_{25}$ concentrations in surface sediments from the Bellingshausen Sea.

the occurrence of sedimentary IPSO$_{25}$ likely signifies the past presence of landfast ice during late spring/early summer. Such an interpretation parallels, to some extent, that for IP$_{25}$ in

the Arctic, the sedimentary presence of which is interpreted as reflecting spring sea ice cover[19,20], although this is not restricted to landfast ice.

Previous palaeo sea ice reconstructions based on IPSO$_{25}$ have largely been conducted on sediment cores retrieved from near-coastal locations around the Antarctic Peninsula (AP)[13,14,16], Adélie Land[11,18] and Prydz Bay[10], so its occurrence in these is consistent with the source and surface sediment data presented herein. Interestingly, however, *B. adeliensis* was not reported in the taxonomic inventories of any of these previous studies, which presumably reflects the susceptibility of this species towards dissolution in the water column and in sediments. Indeed, it has been suggested previously that many sea ice diatoms such as *B. adeliensis* may be under-represented in Antarctic sediments[48], likely as a result of their only lightly silicified frustules. However, on the basis of readily detectable quantities of IPSO$_{25}$ in surface sediments described here, and in previous downcore investigations[10–18], we infer that there must be at least partial deposition of *B. adeliensis* from the melting sea ice to underlying sediments, even if subsequent dissolution of its silica frustules is significant. By combining the intracellular concentration of IPSO$_{25}$ in *B. adeliensis* determined in the current study with its abundance distribution in surface sediments, we estimate that the corresponding contribution of *B. adeliensis* falls within the range ca. $10^2$–$10^5$ cells per gram, which is lower than typical total diatom cell concentrations in Antarctic sediments (ca. $10^5$–$10^7$ cells per gram)[49,50] and certainly below the 2–3% threshold generally used in taxonomy-based paleoceanographic reconstructions[45,51]. On the other hand, the occurrence of IPSO$_{25}$ likely offers complementary information to taxonomic-based sea ice reconstructions, not least, since a clear distinction can be made between the respective signatures of IPSO$_{25}$ and commonly employed sea ice diatom taxa in sediments. Thus, IPSO$_{25}$ is biosynthesized by at least one individual diatom species (*B. adeliensis*) that resides and blooms within the sea ice matrix itself, and therefore represents a proxy measure of late spring/summer (permanent) sea ice. In contrast, the frequently used diatom taxa for palaeo sea ice reconstruction (for example, *Fragilariopsis curta* and *F. cylindrus*[45,46,51]) bloom within the open waters of the marginal ice zone[52,53], and their sedimentary distribution corresponds to seasonal (winter) sea ice cover.

In previous studies, semi-quantitative interpretations of temporal changes to sea ice conditions have been inferred from the variability of IPSO$_{25}$ in downcore records in much the same way that distributions of IP$_{25}$ has been interpreted in Arctic sea ice reconstructions[20]. However, in contrast to IP$_{25}$, there have been, as yet, no published reports that calibrate sedimentary IPSO$_{25}$ concentrations with known sea ice conditions, including seasonal sea ice concentration. Further studies are therefore required to place the interpretations of abundance changes of IPSO$_{25}$ on a firmer footing. In practice, however, the greater heterogeneity of Antarctic sea ice may limit the feasibility of performing such calibrations, especially for coastal locations, owing to the myriad of ice types that exist, the prevalence of polynyas, and the fluctuations that occur within these, both seasonally and annually. In the meantime, we note substantial (orders of magnitude) abundance changes in IPSO$_{25}$, even for sediments from proximate locations within each of the AP, the Weddell Sea and Ross Sea. For example, the concentration ranges of IPSO$_{25}$ are 34–750, 150–1,200, 37–1,100 and 8–1,800 ng g$^{-1}$ in Marguerite Bay (AP), the NW and SE Weddell Sea, and the Ross Sea, respectively.

We suggest that a number of features other than sea ice concentration and/or duration likely have influence over IPSO$_{25}$ abundances and consideration of the ecology of *B. adeliensis* may

offer particularly useful insights. For example, unlike the source diatoms of $IP_{25}$ in Arctic sea ice, which have a generally consistent contribution of ca. 1–5% of the total diatom taxa[21], the proportion of *B. adeliensis* in Antarctic sea ice diatom communities is highly variable, both spatially and temporally, with seasonal shifts in composition also having been observed in previous studies[34,36]. *B. adeliensis* represented the major taxon in our sea ice samples from the WAP, but much lower percentages (and absences) have been reported from the same[54] and other regions[35,36,53], and inter-annual variability is also high. As such, sedimentary $IPSO_{25}$ may be much more strongly influenced by variations in Antarctic sea ice diatom assemblages than is the case for $IP_{25}$ in the Arctic.

*B. adeliensis* is also known to flourish in platelet ice, where diatom communities, in general, can proliferate to yield significantly elevated biomass compared with consolidated ice or other sea ice forms[55–57]. In addition, larger and chain-forming sea ice diatoms such as *Porosira pseudodenticulata* are known to have higher sinking rates compared with solitary pennate species in the Antarctic[58], a phenomenon commonly associated with aggregated sea ice diatoms[59]. The seemingly effective transfer of *B. adeliensis* from sea ice to the underlying sediments, despite its general fragility towards dissolution may, therefore, reflect its propensity to form elongated strands and mats within platelet ice that have elevated sinking rates. Abundances of *B. adeliensis* (and thus $IPSO_{25}$) may thus potentially provide a sensitive proxy indicator of platelet ice, whose occurrence and concentration is commonly associated with the provision of super-cooled low-density sub-surface water derived from nearby ice shelves[55,60–62]. Certainly, the widespread occurrence of *B. adeliensis* and $IPSO_{25}$, with elevated abundances of both for coastal locations, is consistent with ice shelves occupying almost half of the Antarctic coastline and enhanced platelet ice formation within the trajectory of super-cooled surface waters[62].

The production of $IPSO_{25}$ by *B. adeliensis* is not restricted to platelet ice, however, since both have been observed in samples of bottom ice, including those analysed as part of the current study. In fact, production of $IPSO_{25}$ in both platelet ice and consolidated bottom ice communities may potentially explain some of the differences in one of the frequently cited characteristics of $IPSO_{25}$ in Antarctic settings, namely, its relatively enriched $^{13}C$ content (compared with pelagic organic carbon), a feature found previously for other individual lipids and bulk organic matter derived within sea ice[57,63–67]. Although the causes of the enrichment of sea ice-derived particulate organic matter are still debated, $CO_2$ limitation represents the general consensus view, which effectively partially reverses the photosynthetic preference for $^{12}C$ assimilation pertinent to $CO_2$-replete, open water environments, resulting in increased (less negative) $\delta^{13}C$ values[57,63–67]. Indeed, consistent with lower $CO_2$ concentrations in the brine channels of sea ice[68], it has been shown that Antarctic sea ice particulate organic matter can become increasingly enriched in $^{13}C$ with greater distance from the ice/water interface[65], and with the seasonal transition from late winter (October) to spring (December), concomitant with increasing biomass[63], as $CO_2$ potentially becomes depleted. It has also been demonstrated that interstitial $CO_2$ drawdown has a significant impact on $\delta^{13}C$ for particulate organic carbon (POC)[66], although the precise relationship between $\delta^{13}C$ and $CO_2$ supply/demand is probably complicated by variable (and inconsistent) nutrient exchange[67], leading to large ranges in $\delta^{13}C$ for sea ice-derived POC[57,63–67], with only relatively minor $^{13}C$ enrichment in some cases[57,63,67]. In accordance with these previous studies, although some $\delta^{13}C$ data for $IPSO_{25}$ are consistent with significant enrichment of $^{13}C$ within a relatively enclosed (and potentially $CO_2$-limited) sea ice matrix, with $\delta^{13}C = -5.7$ to $-8.5$‰ in sea

ice[12] and ca. $-9$‰ in Ellis Fjord sediments[24], a relatively depleted value ($\delta^{13}C = -17.8$‰) was found for $IPSO_{25}$ in surface sediment material from Adélie land, East Antarctica[12]. Massé *et al.*[12] suggested that this might be attributed to the formation of $IPSO_{25}$ at a time of sea ice melt, when the ice matrix would probably have been more permeable to $CO_2$ and nutrient replenishment. However, given our new findings, we suggest that a more likely explanation for the lighter isotopic composition of $IPSO_{25}$ in Adélie Land sediment is biosynthesis of this biomarker by *B. adeliensis* in the relatively open channels of platelet ice, with comparably free supplementation of $CO_2$ from surrounding waters. An alternative suggestion, involving production of $IPSO_{25}$ in $CO_2$-replete open waters following ice melt seems unlikely given the sensitivity of *B. adeliensis* towards dissolution in the water column. Consistent with our suggestion, Thomas *et al.*[57] found only a small $^{13}C$ enrichment in POC in the interstitial waters of platelet ice from the eastern Weddell Sea (highest $\delta^{13}C = -20.9$‰) with a mean $\delta^{13}C$ value ($-24.0$‰) only slightly higher than that for the underlying open water ($\delta^{13}C = -25.6$‰). On the other hand, a relative enrichment in $^{13}C$ for $IPSO_{25}$ in sediments from Ellis Fjord[24], likely reflects its biosynthesis in the semi-enclosed environment of consolidated bottom ice, which *B. adeliensis* is known to colonize for this region[35]. Within our own samples, we note that $\delta^{13}C$ values for $IPSO_{25}$ in selected surface sediments from the WAP, the northern AP and the SE Weddell Sea, were in the range ca. $-13.5$ to $-15$‰ (that is, intermediate between typical sea ice and open water values), and thus indicative of a semi-enclosed sea ice host. Interestingly, $IPSO_{25}$ abundances in these samples were also among the highest across all samples, consistent with the higher biomass generally associated with platelet ice[55–57].

By further consideration of the environmental sensitivity of *B. adeliensis* (and thus of $IPSO_{25}$) towards nearby glacial and ice shelf water inflow, we provide potentially new insights into the impacts of past changes to such processes by re-inspection of some previous palaeo sea ice reconstructions based on $IPSO_{25}$ (termed diene II in previous reports). For example, we note that enhanced $IPSO_{25}$ in sediment core JPC24 from Prydz Bay (East Antarctica) between ca. 10.9 and 10.4 cal. kyr BP (Fig. 5d), previously interpreted as reflecting heavier sea ice conditions during the deglaciation at this site, also coincided with an interval where the retreating Amery Ice Shelf was probably in the vicinity of, but not over, the core site[10]. Interestingly, lower $IPSO_{25}$ concentrations were observed both before this interval, likely reflecting an ice shelf covering the site, and afterwards, indicative of a more retreated ice shelf edge, as deduced previously, based largely on taxonomic distributions[10]. Similarly, elevated $IPSO_{25}$ in a high resolution (sub-decadal) record from the WAP since ca. 1950 AD (core MTC 18A; Fig. 5b), likely reflects enhanced meltwater-induced platelet ice formation during the recent (and abrupt) retreat of ice shelves and glaciers around the WAP[69,70], rather than an increase in sea ice *per se*. Finally, as part of a Holocene palaeoclimate record for the WAP, Etourneau *et al.*[16] interpreted elevated concentrations of $IPSO_{25}$ in a core from Palmer Deep (core JPC10; Fig. 5e) during the late Holocene (last ca. 3 kyr) as an indication of increased sea ice presence and duration compared with the early and mid-Holocene, stating that glacial ice probably only had a relatively minor influence on $IPSO_{25}$ abundance. An apparent paradox between increased sea ice and coeval increases to surface temperatures within the same record, and enhanced glacial ice melt derived from a nearby core site (core ODP1098; Fig. 5f)[70], was reconciled by proposition of a late Holocene transition towards colder winter/spring seasons and warmer summers. On the other hand, Pike[71] questioned the possible influence of other ice sources on the production and distribution of biomarkers such as $IPSO_{25}$, and we believe that the

new data help resolve this. Thus, we suggest that, enhanced IPSO$_{25}$, observed especially after ca. 3 kyr in Palmer Deep (JPC10; Fig. 5e)[16], likely reflects the positive influence of increasing meltwater discharge from neighbouring ice shelves and glaciers (ODP1098; Fig. 5f)[70], on platelet sea ice formation, with further

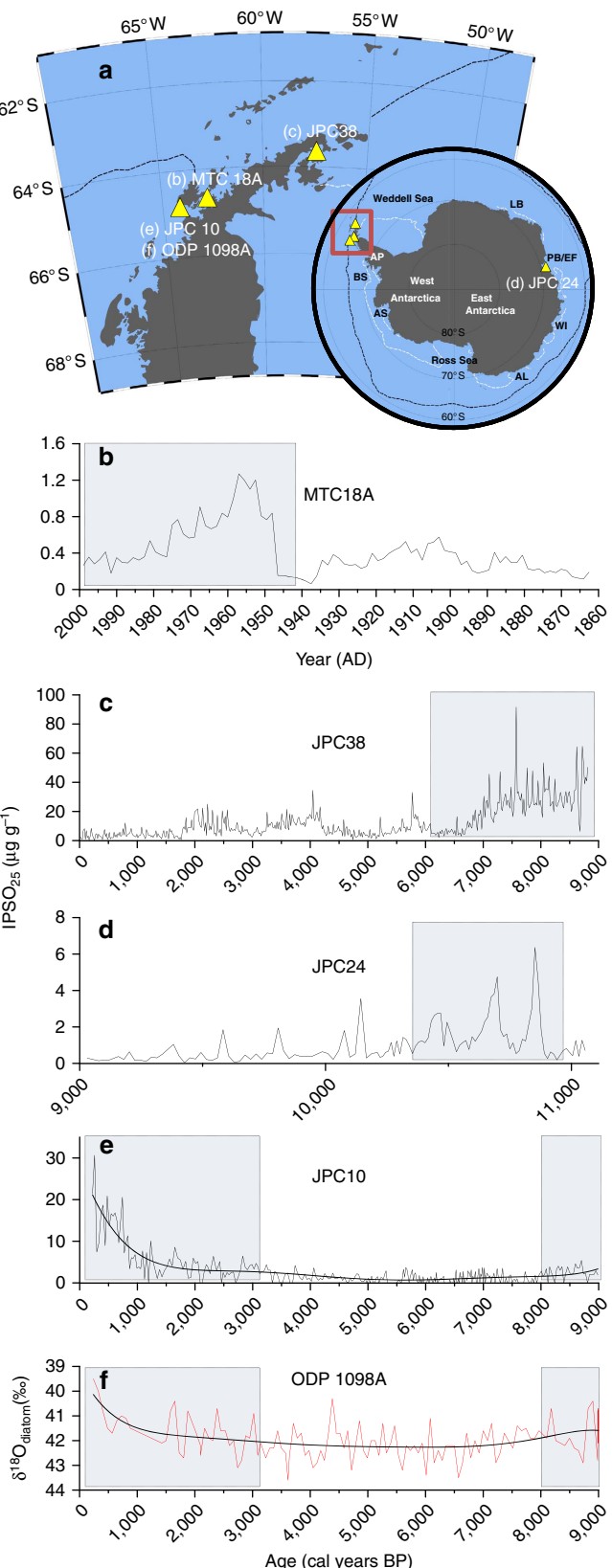

(slightly) elevated concentrations of IPSO$_{25}$ in the earliest part of the JPC10 record (ca. 8–9 kyr BP; Fig. 5e), and also in core JPC38 from the northeastern AP[14] (Fig. 5c), coincident with enhanced glacier discharge or accelerated glacier/iceberg melt during the final stages of the deglaciation[70,72].

Finally, although most of the previous Antarctic studies incorporating IPSO$_{25}$ data have been conducted on near-coastal environments[10,11,13,14,16–18], exceptionally, Collins *et al.*[15] investigated the broader potential of this biomarker for palaeo sea ice reconstruction by determining its distribution in glacial age sediments from three sites in the northern, central and southern sectors of the Scotia Sea, all of which are located further north of the boundary between IPSO$_{25}$ presence/absence in modern surface sediments (Fig. 3). The occurrence of IPSO$_{25}$ in sediments from all three core locations, especially during MIS 2 (ca. 12–24 cal. kyr BP), during which time, sea ice diatom indicator species (that is, *F. curta* and *F. cylindrus*) were also relatively abundant, either suggests that a different source may be responsible for IPSO$_{25}$ biosynthesis in such settings, or that *B. adeliensis* is also able to colonize sea ice for non-coastal marine settings, especially for those under the influence of nearby glacial meltwater.

In conclusion, the identification of the organic geochemical biomarker IPSO$_{25}$ in the Antarctic sea ice diatom *B. adeliensis* likely ensures that future interpretations of the sedimentary occurrence of this sea ice proxy can be made with greater confidence and in more detail. Thus, in addition to representing a qualitative measure of the past occurrence of Antarctic landfast ice during late spring/summer, our findings indicate that variability in sedimentary IPSO$_{25}$ potentially provides further insights into changes to ice shelf and glacial melt processes in long-term records. Further, the determination of the stable isotopic composition ($\delta^{13}$C) may also be particularly enlightening for determining the structural characteristics of the sea ice from which IPSO$_{25}$ was derived (that is, platelet versus consolidated bottom ice). However, measuring the IPSO$_{25}$ content in a larger number of sea ice diatoms and in sea ice of different types and from other regions is required before the interpretation of its precise sedimentary signature can be fully deciphered.

## Methods

**Sample description.** Two landfast sea ice cores were collected from Ryder Bay, situated close to the British Antarctic Survey Rothera Research Station (Adelaide Island; 67°35'8" S, 68°7'59" W) on 3 December 2014. Bottom sections (ca. 10 cm) of each core were sliced from the main cores and left to melt in the dark at 4 °C. Aliquots of the thawed samples were then filtered (GF/F; 0.7 µm), with the remaining material re-frozen and stored at − 20 °C until further use. Surface sediment material from the Antarctic Peninsula, the Bellingshausen Sea, the Amundsen Sea and the Weddell Sea was taken from the upper 0–1 cm of archived box cores, multi-cores and gravity cores held at the British Antarctic Survey, the British Ocean Sediment Core Research Facility (BOSCORF, UK) and the Alfred Wegener Institute for Polar Marine Research (AWI, Germany). Additional sediment samples from the AP and the Ross Sea were collected during several R/V

**Figure 5 | IPSO$_{25}$ in downcore records.** (**a**) Summary map showing the locations and core names where IPSO$_{25}$ (diene II) has been reported in previous studies (note: location (**f**) is a meltwater record only). (**b**–**e**): Abundances of IPSO$_{25}$ in previously published downcore records covering different timescales. The relationship between enhanced IPSO$_{25}$ and increased meltwater inflow from nearby ice shelves and glaciers is highlighted in each case with a shaded blue box. (**b**) WAP[13]; (**c**) northeastern AP[14]; (**d**) Prydz Bay, East Antarctica[10]; (**e**) Palmer Deep (WAP)[16]. (**f**) Meltwater record from core ODP1098 (ref. 70), which is at the same site as JPC10. The smoothed lines in **e** and **f** were created using a sixth-order polynomial function.

ARAON cruises between 2001 and 2015. All surface sediment material was assumed to represent accumulation during the modern era.

**Species identification.** The tube-dwelling *Berkeleya adeliensis* Medlin was identified using scanning electron microscopy (SEM) at the University of Plymouth following the detailed description of Medlin[27]. Preparation of cells for analysis by SEM was carried out by digestion of organic material (HCl; 30 min; 70 °C) followed by washing of frustules ($3 \times 4$ ml $H_2O$). Cleaned frustules were then dried onto glass and sputter-coated (Cr) prior to observation (JEOL 7001F SEM). Morphometric assessment, including elongated central area, apical and transapical length (36 and 8 µm, respectively), number of parallel striae in 10 µm (50) and asymmetric axial area adopting a distinct urn-shape around the simple helictoglossa, support our identification. Individual diatom cells for isolation were identified using a Nikon TS2000 inverted light microscope ($\times 10$ and $\times 40$ objectives) in phase contrast and isolated manually using a modified Pasteur pipette[21].

**Extraction and analysis of lipids.** At the University of Plymouth, HBI lipids were extracted from thawed and filtered sea ice samples, picked cells of *B. adeliensis* and freeze-dried sediments, using previously published methods[21,25,73]. For sea ice samples, lipids were extracted from freeze-dried filters by saponification (20% KOH; 80 °C; 60 min) and then re-extracted with hexane[21]. For isolated cells of *B. adeliensis*, a total hexane extract only was obtained (hexane; $3 \times 1$ ml, ultrasonication; $3 \times 5$ min). In each case, the resulting total hexane extract suspensions were filtered through pre-extracted (dichloromethane/methanol) cotton wool to remove cells before being partially dried ($N_2$ stream) and fractionated into non-polar lipids by column chromatography (hexane (3 ml)/$SiO_2$). For sediments, ca. 1 g of freeze-dried sediment material was extracted by sonication (dichloromethane/methanol; 2:1 v/v, $3 \times 3$ ml) to obtain a total organic extract. Each total organic extract was partially purified to remove polar components, elemental sulphur and saturated non-polar components using silver-ion chromatography[73]. For all sample types, an internal standard (9-octylheptadec-8-ene; 0.01–0.1 µg) was added prior to extraction, to enable subsequent quantification of HBIs by GC–MS methods. Analysis of partially purified non-polar lipids was carried out using GC–MS[73] with identification of $IPSO_{25}$ achieved by comparison of its retention index and mass spectrum with those obtained from a purified standard[23]. Quantification of $IPSO_{25}$ was achieved by integrating individual ion ($m/z$ 348.3) responses in single-ion monitoring mode, and normalizing these to the corresponding peak area of the internal standard and an instrumental response factor derived from a purified standard[23]. GC–MS-derived masses of $IPSO_{25}$ were converted to sedimentary concentrations using the mass of sediment extracted, while a cellular concentration of $IPSO_{25}$ in *B. adeliensis* was obtained by dividing the normalized GC–MS response obtained from picked cells by the number of cells extracted.

**Stable isotope determinations.** The stable (carbon) isotopic composition ($\delta^{13}C$) of $IPSO_{25}$ was determined using gas chromatography–isotope ratio mass spectrometry (GC–IRMS) at Isoprime Ltd, Cheadle Hulme, UK. All GC–IRMS measurements were performed using an IsoPrime100 IRMS with GC5 interface and Agilent 7890B GC installed with an Agilent HP-5MS column (30 m × 0.2 mm internal diameter, film thickness 0.25 µm). All samples were dissolved in hexane (10–150 µl) and injected into splitless mode with the following inlet conditions: 250 °C, purge flow 25 ml min$^{-1}$, purge time 0.75 min. GC carrier gas (He) flow rate was 1 ml min$^{-1}$, oven program as follows: 1 min hold at 50 °C, ramp to 310 °C at 10 °C min$^{-1}$, then 13 min hold. The combustion furnace consisted of a 0.7 mm inner diameter quartz tube packed with CuO pellets, held at 850 °C. GC–IRMS results were calibrated using the certified Indiana alkane standard mix A5 (Indiana University, Bloomington, IN, USA) and all results are thus reported in delta notation ($\delta^{13}C$) relative to VPDB. $IPSO_{25}$ was identified in GC–IRMS chromatograms by retention time comparison with corresponding GC–MS analyses. IonOS software (Isoprime Ltd) was used to process GC–IRMS data; 'peak mapping' functionality was used to systematically designate specific compound identifications across multiple injections for robust data collation. The A5 alkane mix was analysed between every six sample injections in at least duplicate, with calibrations constructed from at least three interspersed measurements of the A5 mix. Reproducibility of individual alkanes was always ≤0.35‰. Root mean standard error (RMSE) of each of the calibrations was usually ≤0.25‰, with an overall RMSE for all calibrations combined of ≤0.21‰, reflecting both the reliability of each calibration, and the long-term stability of the system. Samples containing $IPSO_{25}$ were run in triplicate.

**Data presentation.** Distributions of $IPSO_{25}$ concentrations and study locations referred to in the text are displayed in Figs 1 and 3–5 using Ocean Data View (ODV) software[74].

**Data availability.** The data that support the findings of this study are available from the corresponding author (S.T.B.) and within the article and its Supplementary Information files.

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

## Acknowledgements

This work was supported by the University of Plymouth and the Korea Polar Research Institute (KOPRI, PD15010). We thank Mairi Fenton (BAS) for carrying out the sea ice sampling, Suzanne Maclachlan (British Ocean Sediment Core Research Facility, BOS-CORF, UK) and Rainer Gersonde (AWI, Germany) for sending us with some of the surface sediment material described in this study, and Xavier Crosta (EPOC, France) for providing us with previously reported diene II data. We also acknowledge Peter Bond from the University of Plymouth Electron Microscopy Centre for assistance with SEM and Robert Berstan from Isoprime Limited for assistance with GC–IRMS analysis.

## Author contributions

S.T.B. coordinated the study and wrote the paper. L.S., J.-H.K. and J.-K.G. performed the sedimentary biomarker analyses. T.A.B. conducted the analysis of sea ice samples and the diatom identification. J.-H.K., J.-K.G., J.I.L. and K.-H.S. were responsible for

co-organizing the R/V ARAON cruises and collection of some of the surface sediment samples from the AP and the Ross Sea. K.T. carried out the stable isotope determinations. S.T.B., L.S., T.A.B., J.-H.K. and C.S.A. were responsible for the majority of the data interpretation. All authors commented on the manuscript and discussed the data and implications.

## Additional information

**Competing financial interests:** The authors declare no competing financial interests.

**How to cite this article**: Belt, S. T. *et al.* Source identification and distribution reveals the potential of the geochemical Antarctic sea ice proxy IPSO$_{25}$. *Nat. Commun.* 7:12655 doi: 10.1038/ncomms12655 (2016).

