## [Peer review file · Nature Communications]

Reviewers' Comments:

Reviewer #1 (Remarks to the Author)

A. In the manuscript, the authors present a series of new evidence concerning the recently developed highly branched isoprenoids (HBI) diene II lipid - namely the IPSO25 in this study - as proxy for sea ice (paleo) reconstructions in the Southern Ocean. Belt et al. infer that the major source (if not the exclusive one) of the diene along the Antarctic coast is the diatom species *Berkeleya adeliensis*, which mostly bloom in different sea ice types connected to land during the late spring-early summer season. The authors suggest that the IPSO25 is mainly indicative of landfast sea ice induced by the meltwater discharge from glaciers and ice shelves which is supported by the reinterpretation they make of several previously published paleorecords.

B. The new assumptions developed in this article are certainly of primary importance for a better understanding of the sea ice evolution along the Antarctic margin. Since this relatively new proxy might become a major tool for interpreting paleodata, in a similar way than the IP25 for the Arctic regions, defining the source, distribution and significance is a relevant topic that fits with the expectations of the Nature Communication audience.

C, D and F. Although the reinterpretation of the previous data might be convincing in the light of their new results, before recommending this article for publication, I have some major concerns regarding their new results and questions that would need to be fully addressed by the authors:

1. My first concern is related to the so-called 'IPSO25' as sea ice proxy. I do believe that diene is mostly (uniquely) synthesized below the landfast sea ice by specific diatom species. However, considering strictly the diene, a single molecule, as biomarker, is quite limiting. Comparatively, the IP25 has undergone several modifications before becoming a more robust sea ice proxy in the Arctic. The authors are perfectly aware of that as they took part to its development (Smik et al., 2016). Originally, the IP25 (monoene) was used alone as sea ice biomarker until a more reliable index was developed ($PIP25 = IP25 / (IP25 + \text{phytoplankton biomarker (diene or triene)}) * C$, where $C = \text{mean IP25 concentration} / \text{mean phytoplankton biomarker concentration}$). Here, the authors only mention the diene while the triene for instance, another HBI compound, can be found offshore, in open water conditions, and used (again, similarly to the PIP25) for establishing a better semi-quantitative index (PIP25?). In addition, most of the studies they refer to in the text report a diene/triene ratio. The authors cannot omit it and should thoroughly discuss this issue in their manuscript and explain the reasons why they do consider only diene as sea ice proxy and exclude triene.

2. The name given to this proxy (IPSO25) might not be the most suitable and in my view, too much restrictive. First, for the reasons underlined above: the proxy only includes diene and not the triene or does not correspond to any index. Second, it refers to a proxy for the whole Southern Ocean. It might be true for the glacial periods where the sea ice extended far north but much less for the interglacial periods where the diene is only found along the coast. Again, using an index including the triene would be much more adequate to such name. Third, the nomenclature used is becoming really confusing. In the Arctic, it has been named IP25 (with no specific mention to the Arctic or the monoene lipid) which now becomes PIP25 with the addition of a new parameter (P) corresponding mainly to the 'Arctic' diene or triene which have replaced the use of certain sterols. In the Southern Ocean, IPSO25 refers to the diene (a slightly different molecule) without consideration for the triene or an eventual index. Before adding a new term, a nomenclature should be clearly defined and integrate the possibility of further development which might bring additional names in the future.

3. The authors attribute the diene synthesis to the *B. adeliensis* based on two landfast ice cores

and further data showing that the diene is produced only in the coastal sea ice zone. Although it might be difficult because of the weak preservation of this low silicified diatom species in the underlying sediment, have the authors tried to detect them in the surface sediments used in this study (including SE Weddell Sea and Bellingshausen Sea)? Unfortunately, it seems sediment trap samples are missing for tracking their degradation through the water column. However, how statistically can Belt et al. be confident enough that this species is the unique producer of diene based on two sea ice cores from the Ryder Bay, Western Antarctic Peninsula, for applying it to the whole Antarctica? Have they tried to measure the diene concentration in different picked cells from other species and further offshore?

Some additional comments:

- Lines 37-39: it should be clearly mentioned that sea ice is extending in the Eastern Antarctica, while it is retreating in the Western part (lines 37-39).
- Lines 47-50: the authors should also consider that HBIs, especially diene and triene, might be subject to strong degradation through time and therefore limited for paleoreconstructions beyond the late Quaternary too.
- Line 107: 'firmly' is exaggerated here.
- Line 114: there is no statistical evidence for that.
- Line 186: can it be verified?
- Line 198: this is not completely true. *F. curta* and *F. cylindrus* are used as proxy for winter sea ice extent in case of sediment cores collected offshore. When sediment cores have been taken along the coast, the records based on these two diatom species provide different information: concentration, thickness and seasonality of sea ice. *F. curta* and *F. cylindrus* are mostly linked to fast ice and heavy pack ice. *F. cylindrus* mainly grow during the melting sea ice season, i.e. spring and summer (e.g. Armand et al., 2005) and therefore can trace changes in sea ice during this period. Both diatom assemblages and HBI records are therefore needed to confirm the sea ice trend.
- Lines 343-348: have the surface sediments been dated? Are the authors sure that these samples are modern?
- Line 429: concentration instead of concertation.
- Throughout the text, it should be more readable which samples have been really used in this study for both diene and *B. adeliensis* analyses, and those used from previous studies.

E. Although my comments might look rather negative, the manuscript presents a new exciting result, showing the source of one of the major HBI compound that might be increasingly used in the future for better understanding Antarctic climate. Therefore, if the authors can improve and develop further their hypotheses, this manuscript could really provide interesting insights that corresponds to what we might expect for a high impact journal paper.

G. References are appropriate.

H. The manuscript is well written, clear (except for the study sites that would need some slight improvements to make it more readable), concise. The abstract/summary and conclusions reflect well the content of the article.

Reviewer #2 (Remarks to the Author)

In this manuscript the authors present a case that a geologically important biomarker is produced by a species of diatom commonly found in Antarctic sea ice. If the biomarker is exclusive to sea ice diatoms, then of course it would be a useful proxy for past sea ice extent. However, I do not believe this manuscript demonstrates this link.

The major problems I have with the paper is that although the species of diatom does produce the marker by not testing other common sea ice species there can be no exclusive attribution, and therefore most of the discussion is simply too speculative to be published. Yes this species does

produce the marker, but there has to be a better screening to demonstrate the exclusivity to make the case that the authors do. This is especially true since *B. adeliensis* is only one of many diatoms species common to platelet and land-fast ice systems that they discuss.

Yes the isolated *B. adeliensis* from the ice did produce the marker. However, as evidence for the arguments they then go onto outline in the discussion, the reader does need to have info. from the same species isolated from the water column, and other common sea-ice species isolated from both sea ice and water column. It is difficult to do I appreciate, but just as this team have done for markers in the Arctic, they need more rigorous evidence for this case in the Southern Ocean.

They have plenty of sediment data, but the number of sea ice samples (2) are too small to draw the conclusions that they have. To reach the conclusions that they have, many more sea ice samples, from a much wider distribution in the ice-covered Southern Ocean need to be analyzed.

Considering the paucity of sea ice data, the discussion was very long indeed and too speculative. There is of course scope for interpreting the results and speculating about their consequences. However, I believe that there is not the hard evidence to support the level of speculation in this manuscript.

Reviewer #3 (Remarks to the Author)

This paper presents a set of measurements of a geochemical marker, which the authors name as IPSO25 in sea ice and sediments around Antarctica. This chemical has previously been used in a qualitative way as a palaeo-indicator of sea ice presence, but without the process knowledge that would substantiate such use. Here the authors claim to identify the diatom source, which then allows them to make advances in knowledge of the circumstances in which its presence should be expected. While this somewhat undermines the indiscriminate use made of this marker until now, it does provide a basis for developing its use in the future, albeit in a more limited role, and is therefore of some significance.

The most important result in the paper is the one that identifies the diatom source. This is of sufficiently wide interest to justify publication in *Nature Comms*, and the excellent analytical work is certainly of high quality. However, here I feel the authors need to tighten the logic of the way they describe it. They certainly show that IPSO25 is found in *B. adeliensis*, and with *B. adeliensis* in sea ice. They also identify IPSO25 in sediments particularly under fast ice locations. Thus, as they say, they firmly establish *B. adeliensis* as "a" source of IPSO25, and IPSO25 as the only HBI lipid associated with *B. adeliensis* presence. They even show that related species produce different HBIs. However this does not establish that *B. adeliensis* is "the" only significant source of IPSO25 - to do that requires analysis of sea ice that does NOT contain *B. adeliensis* (showing absence of IPSO25), and/or analysis of all other major diatoms present in Southern Ocean waters (sea-ice- or non-sea-ice-associated) to show that none of them produce IPSO25. This is not done here, and until it is, the authors need to be somewhat more circumspect in assuming that IPSO25 is a sure sign that *B. adeliensis*, and sea ice, were present. My suggestion is that they should moderate some statements, and include a clear description of experiments that could be done (such as those I mention) to nail the issue completely. For example the abstract could read: "The tube-dwelling.....is revealed as a major source of.... (and we make the working hypothesis that it is the main source in Southern Ocean waters and sediments)". This will allow them to proceed while making clear the remaining uncertainty.

The rest of the paper after this first result further discusses the likely significance of IPSO25, and presents some sediment data, attempting to interpret the IPSO25 profiles seen. This is valuable, but is done in a curiously discursive and lengthy way, that seems out of place in a *Nature* family interdisciplinary journal. However, there is nothing especially wrong with it, and I don't propose

that it be excised, merely asking that the authors should themselves review it and see if they could not make it somewhat more attractive to the non-specialist.

Overall the paper does a useful service in defining better the way that IPSO25 might be used and placing its future development on a sound basis. Since establishing palaeo sea ice records is particularly necessary to the enterprise of reconstructing past climate and testing climate models, this is valuable and the paper should be published after relatively minor changes.

Detailed comments:

Abstract, lines 25-29. This is a very long sentence for an abstract, very hard to read. Please split it.

Line 57: This is the first use of the term "Diene II" and is rather curious after you have already introduced the new term IPSO25. I suggest in line 56, in brackets, put "(until now referred to as Diene II, Fig. 1)". You then define and rename it as IPSO25 in line 59.

Line 103-4: "which provides evidence for a strict sea ice origin". Please explain this statement, which is not obvious even for a near-specialist. Why and how does a sea ice origin produce this stable isotope composition, and how does this compare to typical ocean water.

Page 7. I may have missed it but I don't think you ever call Fig 4, which is presumably needed somewhere here.

Line 156 "inspection of surface"

Line 244. This refers to my question related to line 103. Values of -15 are not "enriched" compared to the standard, but I assume you mean they are enriched compared to the ocean as a whole.

Line 289. Amery not Amerys.

Page 13. Please refer in particular parts of this paragraph to particular parts of Fig 5. I found it very confusing because without a great knowledge of Antarctic geography, it's hard to make the link. For example I assume the Prydz Bay core referred to in line 287 is JPC24 and therefore I should look at Fig 5c - this needs spelling out please.

Figure 2. Caption needs work. Firstly part c is not the structure of IPSO25 but is a photo or micrograph. Secondly we need to be told what length the scale bars in each of those is. But also I really don't know what I am supposed to be seeing here. As printed (and most readers will have a similar view), part a may show some diatoms and is OK. Is part b meant to be a single diatom (definition of the scale bar might solve this). And part c is a mystery, it looks like a black field with 2 bubbles on it.

Figs 5 map, the text is too small to read when the figure is printed as a whole page A4 figure.

Responses to reviewers' comments are labeled 'Response' in each case

Reviewer #1 (Remarks to the Author):

A. In the manuscript, the authors present a series of new evidence concerning the recently developed highly branched isoprenoids (HBI) diene II lipid - namely the IPSO25 in this study - as proxy for sea ice (paleo) reconstructions in the Southern Ocean. Belt et al. infer that the major source (if not the exclusive one) of the diene along the Antarctic coast is the diatom species *Berkeleya adeliensis*, which mostly bloom in different sea ice types connected to land during the late spring-early summer season. The authors suggest that the IPSO25 is mainly indicative of landfast sea ice induced by the meltwater discharge from glaciers and ice shelves which is supported by the reinterpretation they make of several previously published paleorecords.

Response: none needed since the reviewer is clear and complementary about the main findings

B. The new assumptions developed in this article are certainly of primary importance for a better understanding of the sea ice evolution along the Antarctic margin. Since this relatively new proxy might become a major tool for interpreting paleodata, in a similar way than the IP25 for the Arctic regions, defining the source, distribution and significance is a relevant topic that fits with the expectations of the Nature Communication audience.

Response: we are pleased that the reviewer believes that our new data align with the expectations of NC.

C, D and F. Although the reinterpretation of the previous data might be convincing in the light of their new results, before recommending this article for publication, I have some major concerns regarding their new results and questions that would need to be fully addressed by the authors:

My first concern is related to the so-called 'IPSO25' as sea ice proxy. I do believe that diene is mostly (uniquely) synthesized below the landfast sea ice by specific diatom species. However, considering strictly the diene, a single molecule, as biomarker, is quite limiting. Comparatively, the IP25 has undergone several modifications before becoming a more robust sea ice proxy in the Arctic. The authors are perfectly aware of that as they took part to its development (Smik et al., 2016). Originally, the IP25 (monoene) was used alone as sea ice biomarker until a more reliable index was developed ($PIP25 = IP25 / (IP25 + \text{phytoplankton biomarker (diene or triene)}) * C$, where $C = \text{mean IP25 concentration} / \text{mean phytoplankton biomarker concentration}$). Here, the authors only mention the diene while the triene for instance, another HBI compound, can be found offshore, in open water conditions, and used (again, similarly to the PIP25) for establishing a better semi-quantitative index (PIPSO25?).

Response: While we follow the general point and understand the potential of the PIP25 index, we believe that the reviewer may not be totally familiar with its current status including its various caveats. Although several papers claim that a combined biomarker approach such as the PIP25 index is more reliable than using a single

biomarker, as far as we are aware, this has yet to be shown to be the case, and certainly not for all regions. Further, PIP25 was 'developed' for the Arctic, not the Antarctic, with sea ice conditions/type, diatom genera, etc, quite different between the two locations. The reviewer is correct to identify a paper by Smik et al that reveals a potential improvement to the PIP25 index, but this is also an Arctic study and for a region that has an extremely regular annual sea ice cycle. It remains to be seen if this new approach works for other regions as stated in the paper. Further, in the Arctic, the PIP25 index has been developed with the aim of better determining sea ice concentration, whereas in the current study, we focus on a different aspect, namely sea ice type or conditions. This is not to say that a PIP25 type index does not have merit for the Antarctic, but this requires a different set of samples and data analysis. We considered presenting other HBI biomarker data within the current study but without their known source (and habitat), we believed this would detract from the main focus of the paper. We also note that, within the Discussion section, we were careful to use previous records of IPSO25 (diene II) only – not any combination with other biomarkers - to illustrate the hypothesized links between sea ice and proximal meltwater processes. Thus, we believe that our discussion is consistent with the new and previous findings.

In addition, most of the studies they refer to in the text report a diene/triene ratio. The authors cannot omit it and should thoroughly discuss this issue in their manuscript and explain the reasons why they do consider only diene as sea ice proxy and exclude triene.

Response: We are aware of the use of the diene/triene ratio in previous studies, although it is unclear to us what the real basis of the use of this ratio is, other than one biomarker is produced in sea ice (diene), while the other is produced by phytoplankton (triene). As far as we are aware, there have, as yet, not been any systematic studies carried out to determine the true significance of the diene/triene ratio, such as a comparison of surface sediment data against known sea ice or other oceanographic conditions. There are also some conflicting interpretations of the diene/triene ratio, which is understandable given current knowledge. Some recent work by Smik and co-authors (Org Geochem, 2016) has shown some possible significance of the D/T ratio but that study also states that much more work is needed.

In contrast, the current study has a clear focus and by identification of its source, we argue IPSO25 is likely influenced by factors quite different from those effecting IP25 in Arctic sea ice. We feel that this is a particularly interesting finding and sets the study apart from one that may have simply established IPSO25 as the IP25 analog for the Antarctic.

We are soon to begin investigating potential sources of the HBI triene, together with its relationship to IPSO25 and sea ice conditions. The outcomes of this investigation will appear in future publications. We could potentially add some comments on this topic but, without any firm data, these would only dilute the impact of the new IPSO25 data presented.

2. The name given to this proxy (IPSO25) might not be the most suitable and in my view, too much restrictive. First, for the reasons underlined above: the proxy only includes diene and not the triene or does not correspond to any index.

Response: We feel that the term follows that commonly employed for the Arctic counterpart – namely a single and signature lipid derived from diatoms dwelling in Arctic (IP25) and Antarctic (IPSO25) sea ice.

Second, it refers to a proxy for the whole Southern Ocean. It might be true for the glacial periods where the sea ice extended far north but much less for the interglacial periods where the diene is only found along the coast.

Response: This is a good point. Previously, this biomarker has generally been referred to as 'diene II' but this is neither intuitive nor necessarily consistent, since there are numerous HBI dienes and 'II' does not confirm anything about the specific structure. Given the new findings, we feel a more informative name is warranted and have proposed the term IPSO25 on the basis of what we now know. It is true that this biomarker may not be present throughout the SO, but understanding the full spectrum of settings is beyond the scope of the current study and like other proxies, will evolve with time. For now, the consistent presence of this lipid in Antarctic sea ice and numerous sediments provides justification for the proposed term. We note that the failure to detect IPSO25 in certain sediments in the current study may, in fact, reflect an analytical LOD rather than a strict absence (as stated in the paper). This aspect is undergoing further investigation.

Again, using an index including the triene would be much more adequate to such name.

Response: Please refer to earlier responses.

Third, the nomenclature used is becoming really confusing. In the Arctic, it has been named IP25 (with no specific mention to the Arctic or the monoene lipid) which now becomes PIP25 with the addition of a new parameter (P) corresponding mainly to the 'Arctic' diene or triene which have replaced the use of certain sterols. In the Southern Ocean, IPSO25 refers to the diene (a slightly different molecule) without consideration for the triene or an eventual index. Before adding a new term, a nomenclature should be clearly defined and integrate the possibility of further development which might bring additional names in the future.

Response: We acknowledge that a number of terms have evolved but we believe there are only three of note:

IP25 – widely recognized as a proxy for Arctic sea ice

PIP25 – a term used for combining IP25 with a phytoplankton marker (P)

IPSO25 – a term to represent a lipid signature of Antarctic Sea ice

We feel that similar (but different) names for lipids produced in Arctic and Antarctic sea ice is a useful feature.

1. The authors attribute the diene synthesis to the *B. adeliensis* based on two landfast ice cores and further data showing that the diene is produced only in the coastal sea ice zone. Although it might be difficult because of the weak preservation of this low

silicified diatom species in the underlying sediment, have the authors tried to detect them in the surface sediments used in this study (including SE Weddell Sea and Bellingshausen Sea)? Unfortunately, it seems sediment trap samples are missing for tracking their degradation through the water column. However, how statistically can Belt et al. be confident enough that this species is the unique producer of diene based on two sea ice cores from the Ryder Bay, Western Antarctic Peninsula, for applying it to the whole Antarctica? Have they tried to measure the diene concentration in different picked cells from other species and further offshore?

Response: We have not been able to analyse other Antarctic sea ice diatom species so far. See more detailed response to this point for Reviewer 3 and in the revised Discussion.

Some additional comments:

- Lines 37-39: it should be clearly mentioned that sea ice is extending in the Eastern Antarctica, while it is retreating in the Western part (lines 37-39).

Response: we understand the point that the reviewer is making, but we believe that, in practice, it is not quite as simple as the reviewer states, hence our careful choice of wording - ...slight overall increase in recent decades, this is not the case for all regions, with dramatic reductions in the Bellingshausen and Amundsen Seas, being of particular note. We do not feel that a location-by-location summary is needed for the generic Introduction.

- Lines 47-50: the authors should also consider that HBIs, especially diene and triene, might be subject to strong degradation through time and therefore limited for paleoreconstructions beyond the late Quaternary too.

Response: While raising the possibility of degradation is a fair point, there have, as yet, been no studies that show that these lipids undergo significant decomposition in sediments, or at least, are more susceptible to degradation than other biomarkers such as IP25. HBI dienes and trienes have been identified in late Pliocene sediments in the Arctic (Stein and Fahl, 2013) and thus far, the same HBIs have been readily identified in last glacial Antarctic sediments (Collins et al., 2013) and in some cases, concentrations of these lipids even increase towards older sediments (see Fig 5). Of course, it remains to be seen what happens for much older sediments. However, the focus of the paper is on a better understanding of what IPSO25 actually represents (when detected).

- Line 107: 'firmly' is exaggerated here.

Response: We agree and have removed this word.

- Line 114: there is no statistical evidence for that.

Response: We have changed the wording to better reflect our findings (see detailed response to Reviewer 3). Note that the section following this revision remains relevant (i.e. further information supporting our suggestion)

- Line 186: can it be verified?

Response: We have not conducted a taxonomic study of the surface sediments, hence we refer to our value range as an estimate.

- Line 198: this is not completely true. *F. curta* and *F. cylindrus* are used as proxy for winter sea ice extent in case of sediment cores collected offshore. When sediment cores have been taken along the coast, the records based on these two diatom species provide different information: concentration, thickness and seasonality of sea ice. *F. curta* and *F. cylindrus* are mostly linked to fast ice and heavy pack ice. *F. cylindrus* mainly grow during the melting sea ice season, i.e. spring and summer (e.g. Armand et al., 2005) and therefore can trace changes in sea ice during this period. Both diatom assemblages and HBI records are therefore needed to confirm the sea ice trend.

*Response: This is a good point and the wording has been changed to better reflect how *F. curta* and *F. cylindrus* are used and what information they yield*

- Lines 343-348: have the surface sediments been dated? Are the authors sure that these samples are modern?

Response: The sediments have not been dated and are assumed to represent modern accumulation. A sentence has been added to explain this.

- Line 429: concentration instead of concertation.

Response: Changed as suggested

- Throughout the text, it should be more readable which samples have been really used in this study for both diene and *B. adeliensis* analyses, and those used from previous studies.

Response: We have checked this aspect and now been explicit when we are referring to previous studies. We also generally refer to our own samples as the ones 'described herein'

E. Although my comments might look rather negative, the manuscript presents a new exciting result, showing the source of one of the major HBI compound that might be increasingly used in the future for better understanding Antarctic climate. Therefore, if the authors can improve and develop further their hypotheses, this manuscript could really provide interesting insights that corresponds to what we might expect for a high impact journal paper.

*Response: We are pleased that the reviewer shares our enthusiasm regarding our findings. We believe that, following some minor revisions (especially within the early Discussion), the source identification (now changed to 'a source') of IPSO25, the consistent occurrence of IPSO25 in surface sediments around coastal Antarctica (not previously shown), the links to the (sea ice) habitat of *B. adeliensis* and the consistency in the IPSO25 response in several previous records to meltwater processes represents such a development of our hypotheses.*

G. References are appropriate.

Response: No response needed.

H. The manuscript is well written, clear (except for the study sites that would need some slight improvements to make it more readable), concise. The abstract/summary and conclusions reflect well the content of the article.

Response: No response needed, except that we acknowledge the complementary feedback.

Reviewer #2 (Remarks to the Author):

In this manuscript the authors present a case that a geologically important biomarker is produced by a species of diatom commonly found in Antarctic sea ice. If the biomarker is exclusive to sea ice diatoms, then of course it would be a useful proxy for past sea ice extent. However, I do not believe this manuscript demonstrates this link.

The major problems I have with the paper is that although the species of diatom does produce the marker by not testing other common sea ice species there can be no exclusive attribution, and therefore most of the discussion is simply too speculative to be published. Yes this species does produce the marker, but there has to be a better screening to demonstrate the exclusivity to make the case that the authors do. This is especially true since *B. adeliensis* is only one of many diatoms species common to platelet and land-fast ice systems that they discuss.

*Response: We have removed any inference that *B. adeliensis* represents the only source of IPSO25. We have also added some further information regarding known HBI producers, which strengthens the argument that *B. adeliensis* is likely a major source. The need to sample further sea ice in the future is, however, noted.*

Yes the isolated *B. adeliensis* from the ice did produce the marker. However, as evidence for the arguments they then go onto outline in the discussion, the reader does need to have info. from the same species isolated from the water column, and other common sea-ice species isolated from both sea ice and water column. It is difficult to do I appreciate, but just as this team have done for markers in the Arctic, they need more rigorous evidence for this case in the Southern Ocean.

Response: The isotopic composition of IPSO25, presented in previous reports and again here, confirms its strict sea ice origin (as explained in the text). As far as we are aware from published data, there have been no reports of IPSO25 produced by open water Antarctic phytoplankton. We acknowledge that there may be other producers of this lipid in sea ice, but please refer to our revised manuscript (and see responses to Reviewer 3) for further discussion of this point.

They have plenty of sediment data, but the number of sea ice samples (2) are too small to draw the conclusions that they have. To reach the conclusions that they have, many more sea ice samples, from a much wider distribution in the ice-covered Southern Ocean need to be analyzed.

Response: We accept the point that further sampling and analysis of more diatoms would strengthen the case and have added this to the text. However, until this is done, we note that IPSO25 and B adeliensis have both been identified in sea ice from other Antarctic regions (as described in the text) and our interpretations are based on the (now) known distribution pattern of IPSO25 and B. adeliensis, together with the known habitat of the latter. The new section regarding potential other sources of IPSO25 also strengthens this point.

Considering the paucity of sea ice data, the discussion was very long indeed and too speculative. There is of course scope for interpreting the results and speculating about their consequences. However, I believe that there is not the hard evidence to support the level of speculation in this manuscript.

Response: Our discussion focuses on a possible reinterpretation of previous HBI diene records. Specifically, it is based on (i) a knowledge of a source of the HBI diene (IPSO25); (ii) the similarity in the sedimentary distribution of IPSO25 with the known ecological habitat of B. adeliensis (landfast ice and platelet ice); (iii) an understanding of the factors that influence the habitat of B. adeliensis (glacial/ice shelf meltwater on platelet ice formation); (iv) temporal synergies between enhanced levels of IPSO25 and known intervals of increased meltwater proximal to the Antarctic coast derived from other proxies.

We believe this approach and interpretation represent a worthwhile alternative to the current view and one that can be tested further in the future.

Reviewer #3 (Remarks to the Author):

This paper presents a set of measurements of a geochemical marker, which the authors name as IPSO25 in sea ice and sediments around Antarctica. This chemical has previously been used in a qualitative way as a palaeo-indicator of sea ice presence, but without the process knowledge that would substantiate such use. Here the authors claim to identify the diatom source, which then allows them to make advances in knowledge of the circumstances in which its presence should be expected. While this somewhat undermines the indiscriminate use made of this marker until now, it does provide a basis for developing its use in the future, albeit in a more limited role, and is therefore of some significance.

Response: we appreciate that the reviewer finds our findings to represent an advance on current (limited) understanding of this proxy

The most important result in the paper is the one that identifies the diatom source. This is of sufficiently wide interest to justify publication in Nature Comms, and the excellent analytical work is certainly of high quality.

Response: we are grateful to this reviewer for this positive and supportive feedback

However, here I feel the authors need to tighten the logic of the way they describe it. They certainly show that IPSO25 is found in B adeliensis, and with B adeliensis in sea ice. They also identify IPSO25 in sediments particularly under fast ice locations. Thus, as they say, they firmly establish B adeliensis as "a" source of IPSO25, and

IPSO₂₅ as the only HBI lipid associated with *B. adeliensis* presence. They even show that related species produce different HBIs. However this does not establish that *B. adeliensis* is "the" only significant source of IPSO₂₅ - to do that requires analysis of sea ice that does NOT contain *B. adeliensis* (showing absence of IPSO₂₅), and/or analysis of all other major diatoms present in Southern Ocean waters (sea-ice- or non-sea-ice-associated) to show that none of them produce IPSO₂₅. This is not done here, and until it is, the authors need to be somewhat more circumspect in assuming that IPSO₂₅ is a sure sign that *B. adeliensis*, and sea ice, were present. My suggestion is that they should moderate some statements, and include a clear description of experiments that could be done (such as those I mention) to nail the issue completely. For example the abstract could read: "The tube-dwelling.....is revealed as a major source of.... (and we make the working hypothesis that it is the main source in Southern Ocean waters and sediments)". This will allow them to proceed while making clear the remaining uncertainty.

This is a good point and we have changed the claims accordingly. See revised Abstract. A new section in the early Discussion regarding known HBI producers and some previous analysis of Antarctic diatom species that have shown an absence of HBIs has also been added. As a result, while we acknowledge the importance of further sampling and analysis of additional species, we describe how:

- (i) *some common and abundant Antarctic sea ice diatom genera (e.g. Fragilariopsis/Chaetoceros/Nitzschia) are known to not produce HBI lipids (Sinninghe Damsté et al., 2004);*
- (ii) *HBIs are only produced by a relatively small number of diatom genera (and only some of these have Antarctic sea ice species (as specified));*
- (iii) *some previous analysis of Antarctic diatoms that are within HBI-producing genera (Navicula glaciei is the major species), did not reveal the presence of any HBIs (including IPSO₂₅). Haslea spp have not been examined but are generally only present in low abundances, unlike B. adeliensis;*
- (iv) *some other potential producers of IPSO₂₅ (e.g. Pleurosigma spp.), do not produce HBIs with the same double bond positions as found for IPSO₂₅.*
- (v) *The observation of only one HBI isomer is rather unusual given the large number of structural types known to be made by other diatoms. Plus, it is much more common for individual diatoms to make several HBIs – only Berkeleya appear to make mainly one. The occurrence of several HBIs in Arctic sea ice is due to the presence of other HBI-producers, for example.*

Collectively, while these points do not entirely eliminate the possibility of there being other IPSO₂₅ producers, there is reasonable evidence to support the claim that B. adeliensis is a major/important source. Some suggestions for further experiments have also been added (see beginning and end of Discussion).

The rest of the paper after this first result further discusses the likely significance of IPSO₂₅, and presents some sediment data, attempting to interpret the IPSO₂₅ profiles seen. This is valuable, but is done in a curiously discursive and lengthy way, that seems out of place in a Nature family interdisciplinary journal. However, there is nothing especially wrong with it, and I don't propose that it be excised, merely asking that the authors should themselves review it and see if they could not make it

somewhat more attractive to the non-specialist.

Response: We feel that, with a novel interpretation of the sedimentary signal of IPSO25, it is worthwhile building the story – see also final response to Reviewer 2. Nature Communications has a significantly larger word limit (and number of references) compared to other Nature journals, which permits more detailed discussions.

Overall the paper does a useful service in defining better the way that IPSO25 might be used and placing its future development on a sound basis. Since establishing palaeo sea ice records is particularly necessary to the enterprise of reconstructing past climate and testing climate models, this is valuable and the paper should be published after relatively minor changes.

Response: we are encouraged that the reviewer believes that our findings are worthy of publication in NC.

Detailed comments:

Abstract, lines 25-29. This is a very long sentence for an abstract, very hard to read. Please split it.

Response: This has been divided into two sentences.

Line 57: This is the first use of the term "Diene II" and is rather curious after you have already introduced the new term IPSO25. I suggest in line 56, in brackets, put "(until now referred to as Diene II, Fig. 1)". You then define and rename it as IPSO25 in line 59.

Response: Changed as suggested. Figure 1 has also been changed with both terms given

Line 103-4: "which provides evidence for a strict sea ice origin". Please explain this statement, which is not obvious even for a near-specialist. Why and how does a sea ice origin produce this stable isotope composition, and how does this compare to typical ocean water.

Response: In fact, this point is discussed in detail in the Discussion section (including our own values), so there is no need to include the interpretation within the Results. As such, this phrase has been removed.

Page 7. I may have missed it but I don't think you ever call Fig 4, which is presumably needed somewhere here.

Response: Good point. This has been added

Line 156 "inspection of surface"

Response: "of" added

Line 244. This refers to my question related to line 103. Values of -15 are not "enriched" compared to the standard, but I assume you mean they are enriched compared to the ocean as a whole.

Response: The sentence has been changed to ... "its relatively enriched ¹³C content (compared to pelagic organic carbon), a feature..."

Line 289. Amery not Amerys.

Response: Changed

Page 13. Please refer in particular parts of this paragraph to particular parts of Fig 5. I found it very confusing because without a great knowledge of Antarctic geography, it's hard to make the link. For example I assume the Prydz Bay core referred to in line 287 is JPC24 and therefore I should look at Fig 5c - this needs spelling out please.

Response: The core names/locations have now been linked in the text with inclusion of core names and individual figure labels (e.g. Fig. 5d)

Figure 2. Caption needs work. Firstly part c is not the structure of IPSO25 but is a photo or micrograph. Secondly we need to be told what length the scale bars in each of those is. But also I really don't know what I am supposed to be seeing here. As printed (and most readers will have a similar view), part a may show some diatoms and is OK. Is part b meant to be a single diatom (definition of the scale bar might solve this). And part c is a mystery, it looks like a black field with 2 bubbles on it.

Response: We have simplified the figure to only show the LM image. The SEM details are given in the Methods

Figs 5 map, the text is too small to read when the figure is printed as a whole page A4 figure.

Response: We have changed the font size to enhance clarity.

Reviewers' Comments:

Reviewer #1 (Remarks to the Author)

Belt et al. present here a revised version of their manuscript entitled 'An organic geochemical proxy for Antarctic sea ice, IPSO25 : source, distribution and environmental significance'. Compared to the previous one, I feel that the changes made by the authors better reflect their main findings but also the current limitations. Although still speculative at some points, due to the limited spatial coverage and the incomplete taxonomic work performed in the sea-ice cores, I believe their study is much more convincing as is and also more honest in a way that the diatom *B. adeliensis* is likely a major source but not the only one, at least until this is confirmed by further work in Antarctic coastal areas, as clearly mentioned by the authors. I still feel that addressing some questions regarding the diene/triene ratio, or diene/triene+diene ratio if developed, would be needed; however, as far as I understand, this will be the focus of another article. This is perfectly understandable that they do not want to dilute the main message here. In sum, I would recommend the publication of this article as is with no specific modifications.

Reviewer #2 (Remarks to the Author)

I still refer back to my original review. As far as I can see there is an awful lot riding on a limited number of sea ice samples (2) and a lack of analyses from other sea ice algae and/or phytoplankton.

The authors have made steps to recognize these facts in the text, but to my mind the study is still at the preliminary findings stage and they have more work to do to be sure of their conclusions that they draw in the extensive discussion, that due to lack of samples has to be speculative (as also invoked by Referee 3).

They now acknowledge that further sampling is needed. I would say that they need to do this before publishing this paper? Nature Communications needs to decide if they want to publish a work based on a very small data set, and as such invoking a significant amount of speculation? Of course future work may show that the authors are correct, and then the risk of publishing now will have been vindicated.

Reviewer #3 (Remarks to the Author)

The authors have responded to my comments with addition of some significant new material substantiating the likelihood that *B. adeliensis* is a/the major source of IPSO25, and also modified their statements to express the remaining uncertainty. They have dealt with my other comments, except for the point about the length of the discussion (also made by another reviewer). I regret that they chose to leave a discussion that is quite challenging for the non-specialist, but I did say that this was an optional change so cannot object now.

Their responses to other reviewers also seem reasonable to me; they have strong (but I think objectively correct) views about PIP25 and the diene/triene ratio, and I think their discussion of these is appropriate.

I am therefore happy now to recommend publication.